# Kidney Transplantation in Older Recipients Regarding Surgical and Clinical Complications, Outcomes, and Survival: A Literature Review

**DOI:** 10.3390/geriatrics9060151

**Published:** 2024-11-20

**Authors:** Aleksandra Barbachowska, Jolanta Gozdowska, Magdalena Durlik

**Affiliations:** Department of Transplantology, Immunology, Nephrology and Internal Medicine, Medical University of Warsaw, 02-006 Warsaw, Poland; ola.barbachowska@gmail.com (A.B.); mdurlik@wum.edu.pl (M.D.)

**Keywords:** older kidney recipients, complications, survival, outcomes

## Abstract

**Context**: The best treatment for end-stage chronic kidney disease (ESKD) is kidney transplantation (KT). As a result of an aging population, each year more kidney transplants in older adults are performed. Nevertheless, older recipients, characterized by more comorbidities and frailty, raise concerns about the outcomes, potential complications, and the general approach. **Aim**: The aim of this literature review was to study the outcomes, graft and patient survival, as well as common complications, to establish safety and increase awareness of the potential complications of kidney transplantation in the older population. **Methods**: PubMed and Google scholar databases were searched. The cut-off age defining an old patient was 60 years. The inclusion criteria were as follows: first kidney transplantation, and studies in English language. The exclusion criteria were as follows: more than one organ transplant, dual transplants, articles published before 2015, meta-analysis, reviews, letter to the editor, case reports, and studies published only as a conference abstract. Comparative and noncomparative studies addressing patient survival, death-censored graft survival, surgical complications, and clinical complications, such as delayed graft function (DGF) and biopsy proven acute rejection (PBAR), were included. **Results**: After screening the papers, 17 studies met the inclusion criteria and were included for review. Eleven papers compared older recipients with younger recipients and in six papers only older patients were analysed. Two studies used paired deceased donors to eliminate donor bias. The rest of the studies used either deceased donors or both living and deceased donors. The majority of patients were male (61.83%) and received a kidney from a deceased donor (58.08%). **Conclusions**: Kidney transplantation is safe and can be beneficial for recipients over 60 years of age. Older patients suffered more infectious complications, which were also one of the main reasons for death. Most studies did not show a significant difference in death-censored graft survival compared to the younger population. More research is needed to establish the prevalence of surgical complications, and some clinical complications.

## 1. Introduction

Kidney transplantation (KT) is by far the most effective way of treating end-stage chronic kidney disease (ESKD) [1]. Compared to dialysis, it prolongs patients’ life and improves quality of life [2,3,4]. Those outcomes also apply to patients over 60 years of age, regardless of donor characteristics [4,5]. Considering the ageing of the population in many countries—for example, in Poland, from 2005 to 2022, the percentage of people over 60 years of age increased from 17.2%, to 25.9% [6]—analogously, a growing number of older patients with ESKD is also observed [7,8]. Thus, the number of older patients on the waiting list, as well as kidney recipients, is constantly increasing [9,10,11]. This group, characterised by more comorbidities, frailty, and often dementia, requires a wide-spectrum and individual approach. This relates not only to which patient should be put on the waiting list, but also how to manage the patient after kidney transplant.

In this literature review, aspects such as post-surgical and clinical complications, mortality, and patient and graft survival after transplantation were considered. The main aim of this review was to establish whether kidney transplant is safe for the older population, and also which aspects are the most important when approaching older patients.

## 2. Methods

In May and June of 2024, PubMed and Google scholar databases were searched using the terms: “older recipients”, “kidney transplant”, “kidney transplantation outcomes”, “clinical complications”, “surgical complications”, and “survival”. In this literature review, the United Nations definition was adopted, thus we were looking for articles that presented patients aged above 60. Other inclusion criteria were first kidney transplantation, and studies in English language. Studies that included more than one organ transplant (multi-organ transplants), dual transplants, or were published before 2015 were excluded. All types of articles were eligible for inclusion except for meta-analysis, review, letter to editor, case report, and studies published only as a conference abstract. The literature search was conducted by a single reviewer. Using the databases mentioned above, the articles were screened by title and abstract to determine if they fulfilled the criteria. Further elimination was made after reading the full text, in regard to the above criteria.

### 2.1. Study and Participant Characteristics

In the presented studies, different cut-off points were established to define older recipients. Eight studies used the age of 60 as the cut-off point, three studies used the age of 65, three studies used the age of 70, one study used the age of 75, and one study used the age of 80. Ten studies compared the older population with younger kidney recipients (excluding paediatric patients). Shi et al. [12] compared older recipients with older, dialysis patients. Six studies were non-comparable and presented outcomes of kidney transplantation in older recipients. Two studies used paired deceased donors to eliminate donor bias. The remaining studies used either deceased donors (four studies) or both living and deceased donors (eleven studies). All studies were retrospective studies and used data obtained from the patients’ medical records. Seven studies were from Europe, four from Asia, three from Australia and New Zealand, two from South America, and one from North America.

### 2.2. Characteristics of Patients

This article reviewed 17 studies, assessing a total of 32,231 recipients, including 5252 older recipients. The majority of patients were male (61.83% of all recipients; 65.91% of older patients). Most patients received a kidney from a deceased donor (58.08% of all recipients; 72.99% of older recipients).

## 3. Analysis of Included Studies

Table 1 presents the characteristics and main findings of the studies used in the literature review.

Gheith et al. studied a total of 962 patients, divided in two groups based on age: one group of 252 patients older than 60 years of age (mean age 65.5), and second group of 710 adults (mean age 49.3). Recipients received kidneys from both deceased and living donors. DDKT predominated in the younger group and LDKT dominated in older patients. PTDM was higher in the younger group (24% vs. 17%, *p* = 0.53); however, diabetes mellitus was more common in the older group. More micro- and macro-angiopathy appeared in the older group. Patients and graft survival were comparable between the older and younger recipients (*p* > 0.5). Older recipients were characterised by much higher cardiovascular risks, and a higher number of malignancies, but fewer episodes of BPAR (*p* < 0.5) [11].

Heldal et al. divided 1326 patients, based on age, into three subgroups (age ≥ 70; age 60–69; age 45–54). Recipients received kidneys from deceased and living donors. No significant difference was observed in death-censored graft survival between all groups (89% in septuagenarians, 88% in seniors; 90% in younger adults). Five-year actuarial patient survival was 56% in the septuagenarians, 72% in the senior group; 91% in the younger group (*p* < 0.001) [13].

In the Doucet et al. study, 299 adults over 70 years of age were compared to 12,684 adults between the ages of 18 and 69. Recipients received kidneys both from deceased and living donors. Older patients had worse one- and five-year survival compared to the younger group (96–97% and 79–81% vs. 97–99% and 90–95%). DGF was comparable between the two groups (*p* = 0.029). Older patients who received a kidney from living donors had lower rates of BPAR (*p* = 0.02) [14].

Orlandi et al. studied a total of 732 patients, divided into two equal groups based on age. The first group included patients over 60 years of age, and the second group had patients between the ages of 18 and 60 years. Diabetes mellitus and prioritisation, but surprisingly not age, were independent risk factors of kidney graft loss. Five-year and ten-year patient survival was lower in the older group (76.6% vs. 87.7% and 54.8% vs. 84.3%; *p* < 0.001). Death with functioning graft was the main reason for graft loss in older recipients. The main causes of death were infections, cardiovascular events, and malignancy. Surgical complications and DGF were more common in the older group. BPAR was similar between the two groups (24.6% vs. 29.5%; *p* = 0.134) [15].

Ko et al. compared a group of 356 patients above the age of 60 with a control group of 4610 younger adults. Recipients received kidneys from deceased and living donors. No significant difference between the two groups was observed in terms of BPAR (*p* = 0.808). Age was an independent risk factor for death-censored graft failure in the overall study population, and in LDKT recipients, but not in DDKT recipients. The main causes of death were once again infections, cardiovascular events, and malignancy. The mortality caused by infection was much higher in the older group than in the control group (*p* < 0.001) [16].

Skrabaka et al. studied 350 patients divided into two equal groups using the same deceased organ donor to establish early post-transplant complications. The cut-off age used in this study was over 60 years and below 60 years. The older group was characterised by a higher body mass index and a higher occurrence of diabetes mellitus and cardiovascular diseases (CVD). In the three-month period studied, no significant difference was observed regarding surgical (20.6% vs. 24%) and clinical complications (28.6% vs. 27.4%), patient survival (95.4% vs. 97.1%), or graft survival (93.1% vs. 95.4%). Age, duration of dialysis, pretransplant diabetes mellitus, and CVD were risk factors for infectious complication in both groups [17].

Yoo et al., in a cohort study, analysed the clinical outcomes of kidney transplantation in the Korean population. A total of 3565 patients, divided into five subgroups based on age (18–29, 30–39, 40–49, 50–59, >60), received grafts from either deceased or living donors. In the older group (age > 60 years) diabetes mellitus and ischemic heart disease were significantly higher compared to the other groups. The recipients’ survival rate was lower in the older group (*p* = 0.001). The main cause of death was infection. BPAR and death-censored graft survival were comparable between all groups (*p* = 0.104; *p* = 0.501) [18].

So et al. examined long-term patients and graft outcomes. Participants over 65 years of age (median age 68 years) received graft from deceased and living donors. One-year and five-year patients’ survival rates were 95.1% and 79%. Moreover, prevalent vascular disease and peritoneal dialysis were risk factors associated with a poorer outcome of KT for older recipients [19].

Adani et al., in a retrospective study, analysed the outcome of KT in recipients over 65 years of age. Two subgroups were selected based on age (65–70; 71–76). All participants received a graft from a deceased donor. Three main causes of death observed during the study were, as previously mentioned, infections, tumours, and cardiovascular diseases. At one, three, five, and ten years, patient survival was 89%, 84%, 72%, and 45%. At one, three, five, and ten years, graft survival was 100%, 97%, 89%, and 84%. Recipients aged ≥ 71 had higher mortality and graft loss compared to the group aged between 65–70 [20].

Saucedo-Crespo et al. compared two groups of patients, defined by an age of greater or equal to 70 years (mean age 73.2 years), and below 70 years (mean age 50.9 years). One-, three-, and five-year patient survival rates were lower in the older group (95% vs. 98%; 86% vs. 95%; 77% vs. 90%; *p* = 0.001). Death was the main factor of graft loss in recipients aged ≥ 70 [21].

Kim et al. studied risk factors and the clinical impact of early post-transplant infection. A total of 3738 patients were divided based on age, with one group equal to or older than 60 years, and another group younger than 60 years. Older recipients had a higher incidence of early post-transplant infections compared to the younger group (22.7% vs. 16.9%), and they more often suffered from mycobacterial infections, coinfections, and multiple site infections. For both groups, bacteria were the most common pathogen causing the infection, and the most frequent site of infection was the urinary tract [22].

In Silva et al.’s study, patients’ short-term survival in 147 recipients aged ≥ 65 years was examined. Patients received kidneys from both deceased and living donors. Factors such as cold ischemia time, increased donor age, cardiovascular disease, DGF, early cardiovascular complication post-KT, early rehospitalizations, and peritoneal dialysis were positively correlated with one-year mortality in older patients [23].

Shi et al., in a matched paired analysis, compared the survival of kidney transplantation with ongoing dialysis for 930 patients over 70 years of age. The research showed that shortly after KT mortality is higher compared to that of people on dialysis. One-year survival rates were similar between the two groups; however, after one year the survival rate was progressively higher in the transplantation group. The main causes of death in both groups were cardiovascular diseases [12].

Neri et al. analysed 452 recipients aged greater than 60 years who received a kidney from a deceased donor. One-, three-, and five-year patients’ survival were 98.7%; 93%; 89%. One-, three-, and five-year grafts’ survival were 94.4%, 87.9%, 81.4%. Age was a risk factor for both patient and graft survival. Moreover, BPAR and neoplasia were correlated with worse graft survival [24].

Cabrera et al. studied older patients with a cut-off age of ≥75 years, which is higher compared to most presented studies. Patients received kidneys from similarly aged, deceased donors. One- and five-year patients’ survival rates, as well as grafts’ survival rates were 82.1%, 60.1%, and 95.6%, 93.1%. In 8.0% of patients, primary graft non-function was observed. Infection was the main cause of death [25].

Jankowska et al. studied 328 patients divided equally into two groups based on age. One group was ≥60 years of age, and the second group was <60 years of age. Patients received a kidney from a paired deceased donor. In the research, no significant difference was observed regarding one-year patient survival, one-year graft survival, DGF, BPAR, or death-censored graft survival between the two groups. Elderly patients had significantly worse patient and graft survival in the long-term. The main causes of death after one year were neoplasia, cardiovascular diseases, and infections [26].

Lønning et al. conducted a study in which two groups of KT recipients were compared in terms of graft and patient survival. All patients received a kidney from deceased donors. The first group was ≥80 years of age and the second group was 70–79 years of age. Contrary to other studies, both groups included the older population. No significant difference was observed regarding death-censored graft survival. The five-year patients’ survival was 55% for recipients ≥80 years of age [27].

## 4. Discussion

### 4.1. Definition of Older Population

The definition of older people remains unclear since there are different cut-off points. The World Health Organization (WHO) defines it as people over 65 years of age (in developed countries). On the other hand, the United Nations points to the age of 60 years. In the medical literature relating to kidney transplants, different cut-off points were also included. For example, in the Middle East Single-Centre retrospective study, the older patients were defined as >60 years old [11], while the study from Norway reported on outcomes in recipients over 70 years of age [13]. The presented studies clearly indicate that kidney transplantation can be beneficial for older recipients; however, some complications—for example, infections—can be observed more frequently in the older population. Thus, clarification in terms of the definition of older kidney recipients is needed.

### 4.2. Surgical Complications

Surgical complications are defined as any incident related to the procedure of surgical transplantation of the graft. Scrabaca et al., in the three-month follow-up period after transplantation, did not observe any difference between older patients and younger recipients [17]. Similar results were observed in a study performed by Jankowska et al. [26]. On the other hand, Orlando et al., in a long-term observational study, proved that older patients have indeed a higher incidence of surgical complications. The most commonly observed were surgical wound dehiscence, urinary fistula, hernia, and dilated bladder [16]. Moreover, the research comparing the Eurotransplant Kidney Allocation System (ETKAS)—a regular program which allocates organs to patients on the waiting list—with the Eurotransplant Senior Program (ESP)—the program in which patients over 65 years of age receive kidneys from deceased donors 65 years and above—indicated that the ESP had a higher rate of surgical complications [28]. Any kind of surgical complication prolongs hospitalisation and also hinder the rehabilitation process, which is essential for older patients. There is a need for more studies about surgical complications, with emphasis on each type of complication.

### 4.3. Clinical Complications

Clinical complications can be described as the ones observed shortly after kidney transplant and in long-term outcomes. In this article, we focus on biopsy-proven acute rejection (BPAR), delayed graft function (DGF), post-transplant diabetes mellitus (PTDM), and infectious complications.

#### 4.3.1. Delayed Graft Function (DGF)

Delayed graft function (DGF) is defined as a requirement for dialysis within seven days after transplantation due to the lack of immediate function of the graft [29]. No difference between the occurrence of DGF in older and younger groups was found in the presented studies [11,14,15,17,26].

#### 4.3.2. Biopsy Proven Acute Rejection

It is clear that, during the ageing process, all body systems, including the immune system, becomes less efficient; therefore, acute rejection in older kidney recipients could be less often observed. However, presented studies are inconclusive in this regard. In a retrospective study performed by Orlandi et al., the incidence of acute rejection in both older and younger recipients was comparable [15]. Similar results were observed in three other studies [13,16,26]. However, Doucet et al. indicated that older patients who received a kidney from living donors had lower rates of acute rejection [14].

#### 4.3.3. Infectious Complication

Infectious complications can be very dangerous, not only shortly after kidney transplantation, but also in the long-term perspective. They are one of the main reasons for death with a functioning graft [13,15,16,18,20,23]. Infectious complications are more often seen in older recipients [22]. The urinary tract is the main site of infection shortly after transplantation [22,23]. The most common pathogens are bacteria. Interestingly, older patients are more frequently exposed to mycobacterial infection, co-infection, and multiple site infections after the first six months post-transplantation [22].

#### 4.3.4. Post-Transplant Diabetes Mellitus

Post-transplant diabetes mellitus has not been a thoroughly examined complication in the presented studies. The reason for this may be the fact that a significant number of older patients already have diabetes mellitus, compared with younger recipients (47.14% vs. 23.3%) [11,14,15,16,17,21,22,27,28]. Only in one study was the number of PTDM higher in older recipients [26]. In contrast to the study by Jankowska et al., the Single-Centre Middle East research observed a higher incidence of PTDM in younger patients, although more micro- and macro-angiopathies were present in the older population [11]. There is no doubt that more research is required to establish whether PTDM is a widespread complication in older patients.

### 4.4. Patent Mortality, Graft Loss, and Patient Survival

As could be presumed, patient mortality in older patients was higher than in younger groups [13,15,18]. The cause of that is not only the age, but also the commonly observed comorbidities in the older population, such as coronary artery disease, diabetes mellitus, and hypertension. Factors associated with a greater risk of all-cause death are prevalent coronary artery disease, cerebrovascular disease, increasing ischemic time, donor age, delayed graft function, and peritoneal dialysis pre-transplantation [19]. The three most common causes of death are infectious diseases, cardiovascular diseases, and cancer [13,15,16,18,20,23,24,25,26]. No difference between death-censored graft survival in various groups was observed [11,13,17,18,26]. Recipient age, BPACR, and surgical complications were identified as risk factors for death-censored graft survival [24]. Regarding patient survival in the three-month follow-up, the older and younger groups were comparable [13,17]. However, the long-term survival varied in favour of younger patients. Orlandi et al. indicated lower survival in the older group after five years (76.6% vs. 87.7%) and ten years (54.8% vs. 84.3%). [15]. The same result was observed in an Australia and New Zealand Dialysis and Transplant Registry Study after a one-year (96% vs. 97%), three-year (84% vs. 93%), and five-year (79% vs. 90%) follow-up [14]. In a Norwegian study, five-year survival for recipients over 80 years of age was 55%, which can still be presumed to be an acceptable outcome [27]. What seems to be an important aspect of patient survival, though, is the comparison between older patients on dialysis, and after kidney transplantation. Shi et al. conducted such a study and the results clearly indicated that, shortly after transplantation, the risk of mortality was indeed higher for KT patients, and this was mainly due to infectious complications. However, after the first year, the survival of KT recipients steadily improved compared to patients on dialysis (five-year survival of 80% vs. 53%, and a ten-year survival of 53% vs. 17%) [12].

### 4.5. Frailty

Frailty is defined as a clinically recognisable state of increased vulnerability resulting from ageing-associated decline in reserve and function across many physiological systems, which leads to an inability to cope with everyday or acute stressors [30]. There are multiple scores used to assess frailty; however, two of them seem to be particularly significant. The first one is the Fried frailty score (phenotype score), and the second one is the frailty score (deficit accumulation score). Fried et al. defined frailty as meeting three or more out of five criteria indicating decreased strength: weakness (low grip strength), low energy, slowed walking speed, low physical activity, and unintentional weight loss (≥4.5 kg in the past year) [30,31]. The deficit accumulation index contains about 50 health-associated deficits, including cognitive function, physical function, functionality, and laboratory test results. The more deficits that are present, the greater the severity of frailty becomes. This scale is also more complex than the phenotype score and requires more laboratory tests [32]. Frailty is an important aspect when approaching older kidney recipients, especially because it is considered a modifiable risk factor and can be treated [33]. It increases mortality and also lowers health-related quality of life [34]. Moreover, it can increase the risk of DGF [35]. In the presented studies, frailty was not widely discussed, thus more studies are needed to establish an accurate approach, not only to kidney transplantation recipients, but also to patients on the waiting list.

### 4.6. Study Limitations

This study has potential limitations. Firstly, its reliance on only two databases, and secondly, the manual verification of excessive amounts of data, make the retrieval process vulnerable to omissions.

## 5. Conclusions

In this literature review, the findings indicate that older kidney transplant recipients have higher mortality, and worse overall survival compared to the younger groups; however, most of the presented studies did not show worse death-censored graft survival. Infectious complications are often observed in older adults and are one of the main causes of death. No difference between the occurrence of DGF and BPAR was observed. More studies are necessary to establish the incidence of surgical complications, as well as some clinical complications, such as post-transplant diabetes mellitus and cytomegalovirus infection. There is also a great need for studies to include frailty and other geriatric impairments occurring in older recipients, which may contribute to some post-transplant complications and affect patient survival. Kidney transplantation can be beneficial for selected older patients; however, each patient requires a careful approach, especially with regard to their comorbidities.

## Figures and Tables

**Table 1 geriatrics-09-00151-t001:** Summary of studies used in the literature review.

Author	Country	Recruitment Period	Recipients Age	Total Number of KT	Donor	Control Group Age (If Presents)	Main Outcomes
Gheith [11]	Kuwait	2000–2014	>60	962	DDKT and LDKT, DDKT predominated in control group and LDKT dominated in older patients	40–60	BPAR was less common for older patients No significant differences regarding patient and graft survival were observed between groups
Heldal [13]	Norway	1990–2005	≥70	1326	DDKT and LDKT	60–69 45–54	No differences in death-censored graft survival between three groups
Doucet [14]	Australia and New Zealand	2000–2015	≥70	10,651	DDKT and LDKT	18–69	Worse patients’ survival and graft outcomes in older group Comparable rates of DGF between groups Lower rates of ABMR in older living donor kidney recipients
Orlandi [15]	Brazil	1998–2010	>60	732	DDKT and LDKT	18–60	Diabetes mellitus was risk factor of kidney graft loss and higher mortality in older patients
Ko [16]	Korea	2009–2012	>60	4966	DDKT and LDKT	<60	KT in older recipients can be associated with worse graft or patient survival High sensitization is less significant impact in older patients Old age is risk factor of higher mortality, mostly due to infection and desensitization therapy
Skrabaka [17]	Poland	1998–2018	>60	350	Paired deceased donor	<60	No difference in regard to surgical and clinical complications between groups No significant difference in patient and graft survival between groups Old age as risk factor of early postoperative infectious complication
Yoo [18]	Korea	1997–2012	>60	3565	DDKT and LDKT	<60	Worse recipients’ survival in older group No difference in death-censored allograft survival between younger and older recipients
So [19]	Australia	2006–2016	≥65	802	DDKT and LDKT	-	Prevalent vascular disease and peritoneal dialysis are risk factors associated with poorer outcome of KT for older recipients 5-year graft and patient survival exceeded 75%
Adani [20]	Italy	1993–2016	>65	109	DDKT	-	KT is safe, feasible, and has good graft survival for older people Recipients age ≥ 71 have higher mortality and higher incidence of graft loss Main causes of death: infectious, tumours, cardiovascular disease
Saucedo-Crespo [21]	USA	2003–2013	≥70	2624	DDKT and LDKT	<70	Acceptable outcome of graft and patient survival in KT recipients age ≥ 70 Caution listing older patient with BMI > 30 kg/m^2^, PRA > 20%, CABG, PVD Death as the main factor of graft loss in recipients age ≥ 70
Kim [22]	Korea	2014–2017	≥60	3738	DDKT and LDKT	<60	older patients had higher incidence of early post-transplant infections older recipients have more mycobacterial infections, coinfections, and multiple site infections
Silva [23]	Portugal	2011–2020	≥65	147	DDKT and LDKT	-	Cautious pretransplant evaluation is needed for older patients Factors such as cold ischemia time, increased donor age, cardiovascular disease, DGF, early cardiovascular complication post KT, early rehospitalizations, peritoneal dialysis, have positive correlation to 1-year mortality in older patients
Shi [12]	Australia	2009–2019	>70	930	DDKT and LDKT		>70 dialysis patients on the waiting list Early post-transplant mortality is higher for older patients compared to patients on dialysis; however, in long term approach, survival of KT recipients is higher
Neri [24]	Italy	2004–2014	>60	452	DDKT	-	Increasing age was risk factor for patient and graft survival BPAR and neoplasia are associate with worse graft survival
Cabrera [25]	Uruguay	2002–2015	≥75	138	Similarly aged DDKT	-	Recipients ≥ 75 years of age can benefit from KT using grafts from extremely aged or deceased donors in comparison to patients remaining on dialysis or listed for transplantation
Jankowska [26]	Poland	1994–2016	≥60	328	Paired deceased donor	<60	No difference in one-year patient survival, one-year graft survival, DGF, BPAR, death-censored graft survival between two groups Older patients have significantly worse patients and graft survival in long-term
Lønning [27]	Norway	1983–2015	≥80	47	DDKT	-	KT from living donor can be beneficial for older recipients 5-year survival rate was 55% for recipients ≥ 80 yo

Abbreviations: DDKT—deceased-donor kidney transplant, LDKT—living-donor kidney transplant, KT—kidney transplant, BPAR—biopsy-proven acute rejection, DGF—delayed graft function, CABG—coronary artery bypass grafting, PVD—peripheral vascular disease.

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
