# Peer review of "Kidney Transplantation in Older Recipients Regarding Surgical and Clinical Complications, Outcomes, and Survival: A Literature Review"

_geriatrics, 2024, doi:10.3390/geriatrics9060151_

Round 1

Reviewer 1 Report (Previous Reviewer 3)

Comments and Suggestions for Authors

I believe the manuscript hav improved sufficiently and my remarks are properly answered. I agree that KT is an option for older ESKD patients, but not all older ESKD patients. I think the authors should add the word "selected" to their conclusion: "Kidney transplantation can be beneficial for selected older patients, however, each patient requires a careful approach, especially with regard to their comorbidities." I also find it odd that several studies included were published before 2015 even though it is stated in the methods that papers published before 2015 was excluded (?). Is it only studies including double transplants or multi-organ transplants published before 2015 that were excluded? The authors also clim that there was only one study (Shi et al) that compared dialysis and transplantation in older patients. In a recent review by Chaudry et al (BMJ 2022 - PMID: 35232772) at least four studies published after 2010 were included. I wonder why these studies were not included in this review?

Author Response

Thank you for all the comments and suggestions. All changes in the manuscript have been marked in green color.

  1. I believe the manuscript hav improved sufficiently and my remarks are properly answered. I agree that KT is an option for older ESKD patients, but not all older ESKD patients. I think the authors should add the word "selected" to their conclusion: "Kidney transplantation can be beneficial for selected older patients, however, each patient requires a careful approach, especially with regard to their comorbidities." 
  •        Thank you for this suggestion. The word “selected” has been added to the text. 

  1. I also find it odd that several studies included were published before 2015 even though it is stated in the methods that papers published before 2015 was excluded (?). 
  • Thank you for this comment. In this review, one of the inclusion criteria was indeed publications after 2015; thus presented studies were published after 2015. However, in the discussion section, we have decided to add some studies published earlier. We thought it would be beneficial for the overall review and would support the results from the main presented studies. 

  1. Is it only studies including double transplants or multi-organ transplants published before 2015 that were excluded? 
  • Both studies including double transplants and multi-organ transplants published before 2015 were excluded. The explanation has been added to the manuscript. 

  1. . The authors also clim that there was only one study (Shi et al) that compared dialysis and transplantation in older patients. In a recent review by Chaudry et al (BMJ 2022 - PMID: 35232772) at least four studies published after 2010 were included. I wonder why these studies were not included in this review?
  • Thank you for this comment. Our study has its limitations; therefore, we added a new paragraph at the end of the discussion section to emphasize that. The databases were searched by a single reviewer; thus, it made the retrieval process vulnerable to omissions.  In the Chaudry et al systematic review, and meta-analysis, six of the included studies were published after 2015. From these six studies only one (Arcos et al study) met the inclusion criteria conducted in our literature review. The other five studies did not meet our criteria (for example, Clark et al included patients after a failed first kidney transplant).

Reviewer 2 Report (Previous Reviewer 1)

Comments and Suggestions for Authors

Overall the paper has vastly improved and reads much better, I would still recommend a limitations section at the end acknowledging the limitations of the literature search and databases used and the fact no quality appraisal was used or described to assess quality of the included papers. You state in the methods section papers that at full text review further papers were eliminated just add some reasons why. You do not have a PRISMA flow diagram which may have helped to demonstrate how studies were included-it is not essential but adds robustness to the methods used and searching. The conclusions needs to start with' In this literature review the findings .... 

In the conclusion section line 312 it would be geriatric impairments not diseases

Author Response

Thank you for all the comments and suggestions. All changes in the manuscript have been marked in green color.

  1. Overall the paper has vastly improved and reads much better, I would still recommend a limitations section at the end acknowledging the limitations of the literature search and databases used and the fact no quality appraisal was used or described to assess quality of the included papers. 
  • A new paragraph regarding the limitations of the study has been added at the end of the discussion section.
  1. You state in the methods section papers that at full text review further papers were eliminated just add some reasons why. 
  • We have added the explanation. Full-text studies were excluded if they did not meet our inclusion criteria. In some articles, the abstract didn’t contain enough information about methodology (for example, if patients received only kidney transplant or whether it was a multi-organ transplant, thus some studies were excluded only after reading the full text). 
  1. You do not have a PRISMA flow diagram which may have helped to demonstrate how studies were included-it is not essential but adds robustness to the methods used and searching. 
  • Since our review is not a systematic review, we have decided not to include the PRISMA flow diagram, to avoid misinterpretation for the readers. However, if necessary, we can prepare such a diagram.
  1. The conclusions need to start with' In this literature review the findings .... 
  • The phrase has been added to the text.
  1. In the conclusion section line 312 it would be geriatric impairments not diseases
  • The word “diseases” has been replaced by the word “impairments”.

This manuscript is a resubmission of an earlier submission. The following is a list of the peer review reports and author responses from that submission.

Round 1

Reviewer 1 Report

Comments and Suggestions for Authors

I would consider the terminology for aging throughout

  • Bowman, C., & Lim, W. M. (2021). How to avoid ageist language in aging research? An overview and guidelines. Activities, Adaptation & Aging45(4), 269–275. https://doi.org/10.1080/01924788.2021.1992712
  • National Institute of Health Blog Don’t Call Me “Old”: Avoiding Ageism When Writing About Aging https://www.nia.nih.gov/research/blog/2023/12/dont-call-me-old-avoiding-ageism-when-writing-about-aging
  • Older People’s Commissioner for Wales Tip Sheet https://olderpeople.wales/wp-content/uploads/2023/07/How_to_avoid_ageism_in_communications_-_Practical_tips_for_professionals.pdf  

Some grammar throughout needs checking line 46-56 check sentences and spelling this needs doing throughout the manuscript there is a spelling mistake in the title as well

I have concerns about the methodology of searching only two databases , very little detail ? There are no limitations discussed.

Add some broader discussion of characteristics of the studies

As the paper is focused on older people I would have thought some mention of other outcomes, frailty and/or other geriatric impairments. There is no wider discussion of other work in this area even in the background where frailty is mentioned.

This paper just needs work on grammar and spelling, addressing the limitations and more detailed methodology and discussion

Comments on the Quality of English Language

The quality of the language is quite poor and needs work

Reviewer 2 Report

Comments and Suggestions for Authors

In summary, this literature review describes an extremely important and highly relevant topic on which further research is urgently needed. The selected literature on which this review is based is highly relevant. Unfortunately, however, the drafting of this review is not yet at an advanced stage. There is a lack of clear structuring throughout the manuscript; introduction, methods, results and discussion are all mixed together, which is very tiring for the reader. Much important information, such as relevant figures, is omitted, and results are not correctly cited, making it unclear on which of the included studies the result is based. The manuscript should be thoroughly revised and resubmitted.

1 Title: Please correct the typing errors in the title.

2 Text: Please avoid the term "elderly" and use "older" instead. 

3 Abstract:

- State inclusion and exclusion criteria

- p.26: higher than what?

- The results are too short and do not provide many results. This part needs to be more detailed to lead to any of the conclusions mentioned.

4 Introduction

- L. 43: Please define frailty.

- Pgs. 47-55: This discussion about which age cut-off to use should be in the methods section.

- The aim of this literature review is not clear and is not linked to the introductory text, therefore the whole introduction to the research question should be reworked.

5 Methods

- L. 60: Please make it clear which search terms were used by separating them with "XXX" and indicate the combinations used.

- Table 1: In the main results you state whether there was a difference or not, but it is not clear what the control group was (healthy? CKD? KT?).

- L. 76: How many percent were male? How many absolute? (for the results!)

- L.74: In the whole results section: If you state that e.g. diabetes mellitus was more common in older patients, then quote from which studies and give numbers (e.g. 50-78% vs. 20-35% control group).

- Characteristics of the patients" please rearrange the whole section and give the most important results first (setting, number of patients, age range (this is completely missing), how many deceased and living donors) and then for example diabetes mellitus. But most of this section belongs in the results section.

- L.83: Please give a quote for this definition.

- L.90: ESP and EKTAS should be explained in the introduction.

- L.92-95: This statement does not belong in the methods section, but in the discussion.

- In the whole manuscript you mix results and discussion and methods and introduction.

Comments on the Quality of English Language

The whole text needs to be proofread by someone who is fluent in English. There are also many typos.

Reviewer 3 Report

Comments and Suggestions for Authors

Thank you for the opportunity to review the manuscript by Barbachowska, Gozdowska and Durlik, submitted for publication in Geriatrics. The manuscript presents a review of papers evaluating kidney transplantation in older recipients, defined as patients older than 60 years. The authors have identified 17 studies. The aim of the study establish whether kidney transplantation in elderly is safe and beneficial, and which aspects are the most important when approaching elderly patients.

To establish knowledge about the benefit of a treatment one should compare with the alternative treatment, which in the case of kidney transplantation is dialysis treatment. Unfortunately, only one of the included studies (ref 22, Shi et al) compared the outcomes with those from a waitlisted dialysis population. Most other included papers are either pure descriptive or compares the outcomes with those of younger kidney transplant recipients. Consequently, the manuscript does not answer the research question. 

The mansucript has more of a description of findings in different studies rather than an analytical approach with discussion of the findings and I don't believe the manuscript add any knowledge about kidney transplantation in the elderly

As more minor remarks there are in some obvious flaws as for example the definition of delayed graft function that the authors define as need of dialysis within 72 hours whereas the common definition is need of dialysis within 1 week.

Comments on the Quality of English Language

The manuscript has several grammatical errors that should be corrected.